# Effect of Mineral Carriers on Biofilm Formation and Nitrogen Removal Activity by an Indigenous Anammox Community from Cold Groundwater Ecosystem Alone and Bioaugmented with Biomass from a “Warm” Anammox Reactor

**DOI:** 10.3390/biology11101421

**Published:** 2022-09-29

**Authors:** Anastasia Vishnyakova, Nadezhda Popova, Grigoriy Artemiev, Ekaterina Botchkova, Yuriy Litti, Alexey Safonov

**Affiliations:** 1Winogradsky Institute of Microbiology, «Fundamentals of Biotechnology» Federal Research Center, Russian Academy of Sciences, 117312 Moscow, Russia; 2Frumkin Institute of Physical Chemistry and Electrochemistry, Russian Academy of Sciences, 119071 Moscow, Russia

**Keywords:** cold anammox, aquifer with extremal nitrogen pollution, permeable barrier, in situ bioremediation, mineral carrier, biofouling, nitrogen removal, bioaugmentation

## Abstract

**Simple Summary:**

During more than 50 years of exploitation of the sludge repositories near Chepetsky Mechanical Plant (Glazov, Udmurtia, Russia) containing solid wastes of uranium and processed polymetallic concentrate, the soluble compounds entered the upper aquifer due to infiltration. Nowadays, this has resulted in a high level of pollution of the groundwater with reduced and oxidized nitrogen compounds. In this work, quartz, kaolin, and bentonite clays from various deposits were shown to induce biofilm formation and enhance nitrogen removal by an indigenous microbial community capable of anaerobic ammonium oxidation with nitrite (anammox) at low temperatures. The addition of a “warm” anammox community was also effective in further improving nitrogen removal and expanding the list of mineral carriers most suitable for creating a permeable reactive barrier. It has been suggested that the anammox activity is determined by the presence of essential trace elements in the carrier, the morphology of its surface, and most importantly, competition from rapidly growing microbial groups. Future work was discussed to adapt the “warm” anammox community to cold and provide the anammox community with nitrite through a partial denitrification route within the scope of sustainable anammox-based bioremediation of a nitrogen-polluted cold aquifer. In this unique habitat, novel species of anammox bacteria that are adapted to cold and heavy nitrogen pollution can be discovered.

**Abstract:**

The complex pollution of aquifers by reduced and oxidized nitrogen compounds is currently considered one of the urgent environmental problems that require non-standard solutions. This work was a laboratory-scale trial to show the feasibility of using various mineral carriers to create a permeable in situ barrier in cold (10 °C) aquifers with extremely high nitrogen pollution and inhabited by the *Candidatus* Scalindua-dominated indigenous anammox community. It has been established that for the removal of ammonium and nitrite in situ due to the predominant contribution of the anammox process, quartz, kaolin clays of the Kantatsky and Kamalinsky deposits, bentonite clay of the Berezovsky deposit, and zeolite of the Kholinsky deposit can be used as components of the permeable barrier. Biofouling of natural loams from a contaminated aquifer can also occur under favorable conditions. It has been suggested that the anammox activity is determined by a number of factors, including the presence of the essential trace elements in the carrier and the surface morphology. However, one of the most important factors is competition with other microbial groups that can develop on the surface of the carrier at a faster rate. For this reason, carriers with a high specific surface area and containing the necessary microelements were overgrown with the most rapidly growing microorganisms. Bioaugmentation with a “warm” anammox community from a laboratory reactor dominated by Ca. Kuenenia improved nitrogen removal rates and biofilm formation on most of the mineral carriers, including bentonite clay of the Dinozavrovoye deposit, as well as loamy rock and zeolite-containing tripoli, in addition to carriers that perform best with the indigenous anammox community. The feasibility of coupled partial denitrification–anammox and the adaptation of a “warm” anammox community to low temperatures and hazardous components contained in polluted groundwater prior to bioaugmentation should be the scope of future research to enhance the anammox process in cold, nitrate-rich aquifers.

## 1. Introduction

The complex pollution of aquatic ecosystems with reduced and oxidized nitrogen compounds is considered one of the urgent ecological problems nowadays. In particular, pollution of the groundwater is one of the important problems requiring non-standard solutions. In situ microbial remediation methods in such ecosystems for the simultaneous removal of ammonium and nitrate are environmentally safe and cost-effective. Among the most promising microorganisms for nitrogen removal in groundwater are anammox bacteria. These autotrophic anaerobes from the phylum *Planctomycetes* use both reduced and oxidized forms of nitrogen (ammonium and nitrite) in their metabolism, forming harmless molecular nitrogen [1]. However, the application of anammox bacteria for bioremediation of the groundwater is complicated due to some physiological features of this group, such as low growth rate, requirements of a certain temperature for growth, and a continuous-flow system with constant substrate supply.

Anammox bacteria have a tendency to the attached growth and formation of multi-species biofilms together with other heterotrophic and autotrophic members of the microbial communities. This property is widely used in wastewater treatment: various biofilm reactor systems show high stability and nitrogen removal efficiency [2,3,4]. One of the promising approaches to creating conditions for the growth of anammox bacteria in aquifers is the application of special inert and semi-inert carriers in the aquifers, similar to those used in bioreactors for wastewater treatment. It was shown previously that the anammox community forms metabolically active biofilms on a wide range of materials that can be used as a permeable biogeochemical barrier. Permeable barriers are a promising form of biogeochemical barrier for groundwater treatment. Such barriers, made of inexpensive materials or the native rock of the water-bearing horizon, do not change the hydrological regimen of the groundwater and provide for macro component removal from the fluid flow [5,6].

One of the main limitations of the wide application of anammox technology for wastewater treatment and bioremediation is the low ambient temperature [7,8,9]. This is especially true for groundwater ecosystems, where the year-round temperature usually does not exceed 10 °C, given that most anammox bacteria exploited for wastewater treatment have a temperature optimum of 30–40 °C [10,11]. It was shown that the specific activity of anammox bacteria can drop dramatically (up to ten times) when the temperature declines from 30 to 10 °C [12], while a sharper activity decline was observed at 15 °C [12,13]. However, numerous works confirm the possibility of the gradual adaptation of the “warm” anammox community to lower temperatures [9]. Moreover, it has been reported that the optimum temperature for anammox activity in natural habitats may be as low as 9 °C in permeable Arctic sediment [14], 12 °C in the arctic, permanently cold marine sediments [15]; or 15 °C in marine sediment from Skaggerak (North Sea) [16]. Physiologically, the low temperature can reduce enzyme activity and the fluidity of bacterial membranes, alter the transport of nutrients and waste products, decrease the rates of transcription, translation, and cell division, denature proteins, fold proteins inappropriately, and form intracellular ice [17]. Thus, the price of adaptation to cold is the conservation of energy for the core metabolism due to a decrease in biosynthesis in general, leading to reduced growth rate and activity of anammox bacteria [9].

For a faster start-up of “cold” anammox or to maintain high anammox activity at low temperatures, it is necessary to maintain a high concentration of anammox bacteria in the system. In addition to the retention of anammox bacteria in the form of granules [18,19] or biofilms [20] on various carriers, the so-called bioaugmentation technology can increase the number of certain microorganisms in the microbial community by adding appropriate exogenous cells or establishing conditions promoting their growth [21,22,23]. In the case of anammox bacteria, bioaugmentation was successfully applied to various wastewater treatment facilities treating landfill leachate [24,25], mainstream [26], and low-strength sewage [27]. This method was beneficial to achieve a faster start-up and increase the efficiency of nitrogen removal [24,25]. Moreover, due to bioaugmentation, anammox bacteria were able to adapt to oxytetracycline that was present in the treated water [28,29] and, presumably, to outcompete nitrite-oxidizing bacteria (NOB) in the bioreactor community [27]. In a low-temperature system, winter bioaugmentation with stored summer sludge helps to restore the anammox activity [30]. Thus, bioaugmentation appears to be an effective tool to increase the abundance and improve the activity of anammox bacteria in various water treating systems and could be potentially used for bioremediation of polluted aquifers as well.

Bioaugmentation technology for bioremediation of polluted underground ecosystems has gained considerable interest. Hazardous compounds, such as carbon tetrachloride, chloroethenes, chlorobenzenes, 1,1,1-trichloroethane, etc., were shown to be successfully removed from the groundwater [31]. In more recent work [32], the feasibility of sequential anaerobic and aerobic bioaugmentation for bioremediation of groundwater contaminated by trichloroethene and 1,4-dioxane was demonstrated. Significant efforts by Michalsen et al. [33,34] were aimed at the use of various strains of the genera *Gordonia* and *Pseudomonas* to clean up a hexahydro-1,3,5-trinitro-1,3,5-triazine (RDX)-contaminated aquifer. Several researchers have also successfully carried out bioremediation of groundwater contaminated with petroleum hydrocarbon [35,36]. It is noted that field-scale trials should be preceded by laboratory evaluation [34].

The research presented herein was undertaken to evaluate the potential for bioremediation of nitrogen-polluted aquifers with an indigenous anammox community adapted to a cold environment. In order to better retain the anammox community in the groundwater system, various mineral carriers were tested to find the best surface and mineral composition for biofilm formation. The “warm” anammox community from the laboratory reactor was also added to the indigenous community to evaluate the possibility of the enhancement of anammox activity in a cold underground environment. To the best of our knowledge, this is the first study of its kind.

## 2. Materials and Methods

### 2.1. Contaminated Groundwater, Containing an Indigenous Anammox Community

Groundwater, contaminated with nitrogen compounds, originated from the water-bearing horizon near Chepetsky Mechanical Plant (CHMZ) (Glazov, Udmurtia, Russia). During more than 50 years of exploitation of the sludge repositories containing solid wastes of uranium and processed polymetallic concentrate, the soluble compounds entered the upper aquifer due to infiltration. Nowadays, this results in a high level of pollution of groundwater by nitrates, ammonium, sulfates, and various metal ions. Technogenic water-bearing horizon was heavily polluted with nitrogen, including reduced and oxidized forms, at a depth of 10–15 m.

Polluted water for the study was taken from a well at a depth of 12 m. The amount of nitrite, presumably of biogenic origin, reached 28 mg/L, providing suitable conditions for anammox bacteria in this unique low-temperature habitat (Table 1). It is important to note the extremely high concentrations of nitrate in the sample, reaching up to 7400 mg/L. Nitrates can be a strong stress factor for the development of microorganisms; on the other hand, they can supply anammox bacteria with necessary nitrite as a result of denitrification. The concentration of bicarbonate ions reached 280 mg/L, which is also an important factor for the development of autotrophic anammox bacteria. One of the dominant contaminants is sulfate ions, which give a prerequisite for the development of ammonium-oxidizing microorganisms due to the processes of sulfate-reducing anaerobic ammonium oxidation [37]. A low content of phosphate indicates a deficiency of this element. The high redox potential of the sample, possibly due to the presence of nitrates, sulfates, and other oxidizing agents, indicates that an intensive anammox process is possible only in microbial biofilms with an Eh gradient or after the removal of nitrate due to denitrification, which contributes to a decrease in the redox potential. The low content of organic matter in the system does not allow for effective self-cleaning of the system from nitrogen due to heterotrophic denitrification. For this reason, the only possible way of self-cleaning the system from ammonium is anaerobic ammonium oxidation in anammox processes or aerobic ammonium oxidation in the nitrification process. The water also contained technogenic uranium at a concentration of 250 µg/L.

Thus, the development of biofilms is necessary for the anammox process under conditions of oxidative and nitrate stress. Composition of the microbial community of the groundwater is discussed below in the Results and Discussion section.

### 2.2. Bioaugmentating Anammox Biomass

Biomass from the laboratory up-flow bioreactor was used as an allochthonous (bioaugmentation) anammox community [22]. The reactor was operated at room temperature (24–28 °C) and fed with synthetic mineral media, containing 200 mg N-NH_4_/L and 200 mg N-NO_2_/L. Anammox biomass formed granules up to 3 mm in diameter. The granules were preliminarily dispersed and passed through a 0.2 mm sieve to obtain a highly dispersed cell suspension.

### 2.3. Mineral Carriers

Two types of substances were used as carriers: mineral materials used as permeable barriers—different silicates and aluminosilicates (vermiculite, zeolite, expanded clay, and sand), and materials of the native rock from the water-bearing horizon (Table 2). For comparison, in the current study, the native rock from another water-bearing horizon, B2, was used, from a depth of 12 m with high levels of nitrogen pollution. The study of in situ biological treatment was carried out previously at this site [38].

The main attention was paid to the clay minerals. Clays are used as components of non-permeable vertical barriers or cutoff slurry walls [39]. Such barriers are used to slow down or stop the movement of the formation water or to create “reactive gates” to direct flow into the treatment zone in a reactive permeable barrier. In the current study, the natural clays of the bentonite and kaolin types from various deposits on the territory of the Russian Federation were used (Appendix A). The composition of the clays was studied in detail previously [40].

The ChMZ rock used in this study was sampled from water-bearing horizons from a depth of 10–11 m in parallel with groundwater sampling. Its mineralogical composition includes predominantly quartz (45–50%), a mixture of clay minerals (kaolinite, illite) (∑ 15–20%), potassium feldspar (20–25%), plagioclase (5–10%), and 5–10% Fe-containing minerals, in total. B2 rock is a loam containing the following substances: quartz (~50%), potassium feldspar (~24%), albite (~5%), and also up to 5% Fe-containing minerals (goethite or siderite). Clay fraction contains smectite (~7 to 8%), illite (~5 to 6%), and kaolinite.

Elemental composition of the mineral carriers used in the study is shown in Table 3. Carriers had different specific surface areas. Bentonite clays and expanded clay had the most developed surface. The minimal specific surface area was typical for sand and vermiculite (it did not exceed 5 m^2^/g).

### 2.4. Experimental Setup

Briefly, 1 g of the mineral carriers (see Table 2) was placed into each sterile 30 mL glass vial together with 20 ml of groundwater. In groundwater, the concentration of ammonium and nitrite nitrogen was preliminarily set at 50 mg N-NH_4_/L and 60 mg N-NO_2_/L by adding the appropriate amount of nitrogen salts (NH_4_Cl and NaNO_2_). Then, 1 mL (pre-washed and diluted with the distilled water) of the bioaugmenting anammox biomass suspension from the laboratory reactor was placed into one-half of the vials. The relative number of anammox cells, according to direct counting of cells hybridized with a Cy-3-labeled anammox-specific FISH probe, in groundwater was 2 × 10^4^ cells/ml, in suspension from the anammox reactor—2 × 10^6^ cells/mL. The glass vials were thoroughly purged with argon and then sealed with a rubber stopper and an aluminum crimp cap. With each of the mineral carriers, the experiment was carried out in duplicate, both for indigenous and bioaugmented anammox communities. Incubation was carried out without stirring in the dark at 10 °C for 35 days. Samples for the determination of ammonia, nitrite, and nitrate nitrogen, pH, and Eh were taken every 5 days.

### 2.5. High-Throughput Sequencing of 16S rRNA Genes

DNA was isolated using the FastDNA™ SPIN Kit for Soil (MP Biomedicals, Santa Ana, CA, USA) according to the manufacturer’s instructions. The preparation of amplicon libraries of the V4 region of the 16S rRNA gene was carried out as described previously [41] using a pair of primers: 515F (5′-GTGBCAGCMGCCGCGGTAA-3′ [42]), Pro-mod-805R (5′-GGACTACHVGGGTWTCTAAT-3′ [43]). The libraries were sequenced on a MiSeq system (Illumina, California, United States) using a 150-nucleotide length paired-end read cartridge. Libraries were prepared and sequenced into two replicas for each sample. The amplicon sequence variant (ASV) table was constructed using the Dada2 script [44] and the SILVA 138.1 database [45]. Analysis of the ASV table was performed using MicrobiomeAnalyst [46].

### 2.6. Analytical Methods

The elemental composition of mineral carriers was determined using X-ray spectral fluorescence analysis on Axios mAX Advanced sequential vacuum spectrometer (with dispersion by the wavelength) (PANalytical, Netherlands). Standard samples of the US Geological Survey (USGS) were used as the controls. Organic carbon (C_org_) was determined by the total organic carbon analyzer Shimadzu TOC-V Series (Japan).

The determination of Eh and pH values in groundwater was carried out using an ANION-4100 pH meter/ionometer (Infraspak-Analyte, Russia) with a combination of electrodes. The pH and Eh during incubation experiments were determined using an FE20 pH meter equipped with an InLab^®^ microelectrode and ORP electrode InLab Redox Micro (all Mettler Toledo, Switzerland).

The concentrations of anions and cations were measured on a Capel-205 new generation capillary electrophoresis system (Lumex, Russia) according to the manufacturer’s recommendations. Electrophoresis was performed in untreated fused-silica capillaries of 60 cm length (effective length, 50 cm) and 75 μm internal diameter. The capillary was held at 20 °C and the applied voltage was +13 kV for cations or −17 kV for anions [6].

Development rates of microbial communities on mineral carriers were determined using the MTT assay [6].

### 2.7. Microscopy

Visualization and analysis of the microbial biofilm on the surface of the mineral carrier were performed using a Leica SP5 confocal scanning laser microscope (Leica, Germany) and program package Comstat 2.1 for ImageJ. Staining of the polysaccharide matrix was performed with lectin concavalin A (conA), conjugated with Alexa Fluor-488 (C11252, Thermo Fisher), for 30 min in dark. After staining, the samples were washed with distilled water and visualized at 100× magnification. The surface of the biofilm was calculated using ImageJ 1.52 with program package COMSTAT 2.0.

Fluorescence in situ hybridization (FISH) was applied to estimate the presence and relative abundance of anammox bacteria in intact samples. For hybridization, 16S rRNA-specific Cy-3-labelled oligonucleotide probe amx368 (provided by “Syntol”, Russia) was used. Hybridization was carried out at a temperature of 46 °C according to the standard procedure [47], with modifications [48,49].

Phase contrast and epifluorescence microscopy were carried out using a phase contrast microscope Leitz (Germany) with a Zeiss 20 filter for Cy3-labeled probes, with a digital camera Nikon DS-Fi1c (Japan) at 100 × 10 magnification.

### 2.8. Calculations and Correlation Analysis

Ammonium and nitrite nitrogen consumption rate v(N) was calculated according to Equation (1):(1)$$v\left(N\right)=\frac{c{\left(N\right)}_{\mathrm{init}.}-c{\left(N\right)}_{\mathrm{fin}.}}{t}$$
where

c(N)_init._ and c(N)_fin._—initial and final concentrations of nitrite or ammonium nitrogen in the incubation medium, mg N/L,

t—incubation time, day.

N-NO_2_/N-NH_4_ index was calculated using Equation (2):(2)$$N-{\mathrm{NO}}_{2}/N-{\mathrm{NH}}_{4}\;\mathrm{index}\;=\frac{c{\left(N-{\mathrm{NO}}_{2}\right)}_{\mathrm{init}.}-c{\left(N-{\mathrm{NO}}_{2}\right)}_{\mathrm{fin}.}}{c{\left(N-{\mathrm{NH}}_{4}\right)}_{\mathrm{init}.}-c{\left(N-{\mathrm{NH}}_{4}\right)}_{\mathrm{fin}.}}$$
where

c(N-NO_2_)_init._ and c(N-NO_2_)_fin._—the initial and final concentrations of nitrite nitrogen in the incubation medium, mg N/L,

c(N-NH_4_)_init._ and c(N-NH_4_)_fin._—the initial and final concentrations of ammonium nitrogen in the incubation medium, mg N/L.

The coefficient (coeff.) of changes in biofouling area and ammonium uptake in the bioaugmentation mode was calculated by dividing the parameter values obtained in the bioaugmentation mode (X_bioaug.)_ by the corresponding parameter values obtained without bioaugmentation (X_indig._), according to Equation (3).
(3)$$\mathrm{coeff}.=\frac{X_{\mathrm{bioaug}.}}{X_{\mathrm{indig}.}}$$

Correlation analysis and graphical visualization of the calculations were carried out using the program Origin Pro 2021b.

## 3. Results

### 3.1. Composition of the Indigenous Anammox Community

The composition of the microbial community based on the 16S rRNA sequencing is represented in Figure 1. The most numerous bacteria in the sample from the aquifer were representatives of the *Pseudomonas* genus of the *Pseudomonadaceae* family. This genus is metabolically versatile and is known for its ability for nitrification, aerobic and anaerobic denitrification, assimilative sulfate reduction, and reduction of heavy metal ions [50,51]. *Pseudomonas* possesses an active quorum sensing system (QS), which is known to regulate biofilm formation and the production of extracellular DNA and large surface proteins and surfactants [52].

Among the *Planctomycetes*, the most abundant was the family *Scalinduaceae*—genus *Candidatus* Scalindua. These bacteria are responsible for the anammox process in the community. Members of this family are typically found in seawater; they are considered to be halophilic bacteria, in some cases, phychrotolerant [53]. Ca. Scalindua was previously reported to inhabit soil and groundwater communities, though its abundance was not as high as that of other anammox genera [53,54,55].

Members of the genus *Pantoea* (family *Erwiniaceae*) are predominantly facultative anaerobes, and are able to combine the oxidation of acetate or H_2_ and the reduction of different electron acceptors including Fe(III), Mn(IV), Cr(VI), and As(V) [56,57].

Members of the genus *Paenibacillus* (family *Paenibacillaceae*) are organotrophic aerobic or facultative anaerobic bacteria. Some of them are capable of metal reduction (for instance, Fe(III)) and uranium immobilization. Presumably, they could be responsible for exopolysaccharide matrix synthesis in biofilm [58,59]. Some representatives of this genus could also be involved in N-cycling processes since they carry out N_2_ fixation [60]. Another important feature is their ability to participate in the solubilization of phosphate-containing minerals, which can promote the growth of anammox bacteria and other microorganisms of the studied community [61].

Members of the *Devosiaceae* family may also be involved in nitrogen transformation in the community. Most of the *Devosiaceae* are organotrophs capable of nitrification [62] and the subsequent reduction of nitrite to nitric oxide during denitrification [63]. Moreover, some members of the genus *Devosia* possess a nirB gene, which is responsible for nitrite reduction to ammonium (*D. insengisoli*, according to the Kegg pathway module https://www.genome.jp/pathway/dea00910 accessed on 23 August 2022).

Most of the known members of the genus *Aeromonas* (family *Aeromonadaceae*) are mesophilic microorganisms performing heterotrophic nitrification and aerobic denitrification at high pH levels. Some of them also reduce iron (III) and carry out assimilative sulfate reduction [64,65].

### 3.2. Bioaugmenting Anammox Community

Figure 2 represents the composition of the microbial community of the anammox reactor, which was described previously [22,66]. The key microbial group in that community was anammox bacteria. The dominant anammox phylotype belonged to the genus Ca. Kuenenia. Members of the genera Ca. Jettenia and Ca. Brocadia represent the minor part of the anammox community. Interestingly, the results obtained demonstrate a change in the composition of anammox bacteria in the community. Thus, according to a previously published metagenomic review, this community was dominated by Ca. Jettenia [66], but then they were replaced by Ca. Kuenenia, which was previously less numerous, together with Ca. Brocadia, accounting for less than 5% each [22].

Phylum *Chloroflexi* (family *Anaerolineaceae*) appears to be one of the most abundant in this community. Members of this group are ubiquitous in such communities and are considered to play an important role in granulation [67,68]. Some of them also could be involved in N-transformation in the bioreactor, together with denitrifying members of the families *Burkholderiaceae* and *Hyphomicrobiaceae,* which are also present in the community [68,69,70,71]. In addition, the community included another important participant of the N-cycling-aerobic ammonia-oxidizing *Nitrosomonas*, which is often abundant in anammox bioreactors and can even co-exist in granules with anammox bacteria [72,73,74]. Members of the phylum *Bacteroidota* (families *Chitinophagaceae*, *Kapabacteriales*, and *Microscillaceae*) are an essential part of the community. Their presence is considered to be important for the functioning of such a community and their abundance is usually high [68,72,75,76]. Most of them are organotrophs, but some are involved in nitrogen reduction processes [73,75].

Other widespread *Planctomycetes* include the family *Phycisphaeraceae*, which are not associated with the anammox process but are responsible for the decomposition of organic matter and have previously been found in the same marine sediment community with anammox bacteria [77,78].

### 3.3. Microscopy of Indigenous and Bioaugmentation Anammox Community

Results of the phase contrast (Figure 3A,C) and fluorescent (Figure 3B,D) microscopy of the indigenous groundwater community and the biomass of the laboratory reactor (bioaugmenting community) demonstrate the presence and activity of anammox bacteria in both communities. Both communities consist of clustered microbial cells of various morphology, mostly cocci, bacilli, and some filamentous cells. Due to the dense clusterization, the quantitative analysis of the hybridized anammox cells was not performed. Anammox cells hybridized with a specific FISH probe formed microcolonies of several cells with a diameter of 5–25 µm. The size of the microcolonies presumably indicates the stage of development of the microcolonies. In the groundwater community, only young (i.e., small) microcolonies were found. The relative number of anammox bacteria in the bioaugmenting community was more than 50% of the total number of microbial cells, while in the indigenous community of groundwater, the relative number of anammox cells was less than 15%.

### 3.4. Effect of Mineral Carriers on the Performance of the Indigenous Anammox Community

The results of measuring the respiratory activity, the area of biofouling, and the rates of nitrogen consumption are shown in Figure 4 and Figure 5. The maximum respiratory activity was observed on clays HB, KS3, and KS1 on zeolite Z1 and illite. Compared to these materials, the respiratory activity on polymineral rocks was significantly lower. Minimal respiratory activity was observed on quartz, zeolite Z2, sand, and vermiculite. At the same time, in our other works, it was shown that the biofouling of vermiculite can be quite high [5,38]. An assessment of the total area of the polysaccharide biofilm showed that minimal biofouling was observed on quartz, sand, zeolite Z2, and KS1; on other materials, biofouling was more than 50%.

The consumption rates of ammonium and nitrite, as well as the ratio of consumed ammonium and nitrite nitrogen (N-NO_2_/N-NH_4_), are shown in Figure 5. According to the canonical equation of the anammox process, N-NO_2_/N-NH_4_ is 1.32 and, according to the updated data, 1.146 [79]. At the same time, it is known that this ratio can vary from 0.8 to 2.0 and higher [80]. Thus, when the N-NO_2_/N-NH_4_ is in this range, we can assume a greater contribution of the anammox process to the total removal of nitrogen in the system, and with a significant deviation from these values, an increase in the contribution of such processes as denitrification or nitrification can be possible.

It has been established that not in all cases a high degree of biofouling correlates with the occurrence of the anammox process. Thus, on quartz with a low biofouling area and low rates of ammonium and nitrite consumption, according to N-NO_2_/N-NH_4_, the process was most likely carried out by anammox bacteria. On ilite (IL), a high biofouling area was observed throughout the experiment, and the process of nitrite and ammonium consumption was of medium intensity, while, according to N-NO_2_/N-NH_4_, it was also probably carried out by anammox bacteria. On kaolin clays (KC1, KC2), a high value of respiratory activity, moderate biofouling, and the rate of consumption of nitrite and ammonium were revealed.

Judging by the ratio N-NO_2_/N-NH_4_, the main removal of nitrogen can be carried out by anammox bacteria. At the same time, on kaolin clay (KC3), the highest fouling area, a high nitrite consumption rate, moderate respiratory activity, and low ammonium consumption rate were observed. According to the N-NO_2_/N-NH_4_, the main process of nitrogen removal carried out was likely due to denitrifying bacteria, with a smaller contribution of anammox bacteria. On bentonite clays BC1 and BC2, a high value of the fouling area and the rate of consumption of nitrite were revealed; anammox bacteria could make a significant contribution to nitrogen removal according to the N-NO_2_/N-NH_4_. On bentonite clay BC3, a high value of the fouling area and the rate of nitrite consumption, a moderate respiratory activity, and a low ammonium removal rate were observed. According to N-NO_2_/N-NH_4_, both anammox and denitrifying bacteria contributed to nitrogen removal.

On activated Khakasskiy bentonite (BC4), a high fouling area and respiratory activity, and moderate rates of ammonium and nitrite consumption were observed. According to the N-NO_2_/N-NH_4_, anammox bacteria may have played a significant role in the process of nitrogen consumption. On the CHMZ rock, a high value of all indicators was found, except for respiratory activity. This is probably due to the fact that anaerobic anammox bacteria developed in this case. On the B2 rock, high values of the fouling area and nitrite consumption rate, and low values of respiratory activity and ammonium consumption rate were found, which is probably due to the predominance of denitrifying microorganisms in the microbial community along with anammox bacteria. On vermiculite (V and expanded clay), a high value of the fouling area and low values of all other indicators were found. On zeolite (Z1), high values of all indicators were revealed. It can be concluded that this material is well suited for the development of bacteria involved in the nitrogen cycle. Presumably, anammox bacteria were also present in this community. Zeolite (Z2) showed low fouling area, respiratory activity, and rate of nitrite and ammonium consumption. According to the N-NO_2_/N-NH_4_ ratio, the presence of anammox bacteria on this carrier was insufficient.

### 3.5. Effect of Bioaugmentation with a “Warm” Anammox Community

The results of carrier biofouling in the bioaugmentation mode are shown in Figure 6, and the main differences from the indigenous anammox community are shown in Table 4.

The addition of the “warm” anammox community from the laboratory reactor led to an increase in nitrogen consumption and the total area of biofouling on IL, KC2, KC3, BC1, BC3, HB, S, and Z2. The greatest increase in NH_4_ consumption was observed on B2 rock (by 7.7 times), then KC2 (by 2.8 times), and EC (by 2.4 times), the rest were less pronounced. Surprisingly, Q, KC1, BC2, and especially CHMZ rock (0.2 times), showed signs of inhibition in the activity of the “warm” anammox community, since ammonium consumption after bioaugmentation was lower than by the indigenous anammox community. A tendency to increase the area of biofouling with a decrease in nitrogen consumption was observed on KC1, BC2, and CHMZ rock. At the same time, on B2 rock, V, and EC, an increase in nitrogen consumption was noted with a decrease in the area of biofouling. It should be noted that in the bioaugmentation mode, according to the ratio of consumed N-NO_2_ to N-NH_4_, the contribution of the anammox process to nitrogen removal was maximal on KC1, BC1, BC3, CHMZ, B2, Z1, and Z2.

## 4. Discussion

The microbial composition of the two communities varied remarkably from each other. Anammox bacteria were present in both communities. Genus Ca. Scalindua, which was found in the groundwater community, is a typical anammox inhabitant of natural ecosystems such as seawater, sediments, and soils [54,55,81]. At the same time, the bioaugmenting community included anammox bacteria of the family *Brocadiaceae,* which is usually associated with anthropogenic habitats. In addition, this community included other microbial participants in the N-cycle and bacteria important for the functioning of wastewater treatment communities and biofilm formation [68,72]. This composition of the microbial community is a result of long-term cultivation under favorable laboratory conditions [18,19,66]. Some of the phylotypes from the groundwater community also could be involved in nitrogen transformations, and some phylotypes are associated with groundwater habitats, including contaminated groundwater [82,83].

Currently, one of the most intensively studied ways to stimulate the growth of biofilms in the anammox community, to increase specific anammox activity or maintain it under adverse environmental conditions, is the application of special carriers. This area of research is still developing, but it is already known that carriers of different natures can have different effects on the anammox process [84,85]. According to our results, zeolite, ChMZ aquifer rock, illite, and kaolin clays turned out to be the most suitable carriers for creating a permeable nitrogen removal barrier. These results are consistent with our previous study in which we used a microbial community from a “warm” anammox reactor and the incubation temperature was high (30 °C) [6]. The intensity of biofouling of mineral carriers by microorganisms depends on a number of factors. One of the most important is the presence of biogenic elements in them and their availability for microorganisms. Another is the features of the surface of the material: morphology, charge, the presence of free functional groups, etc.

To assess the effect of the elemental composition on the process of nitrogen removal and biofilm formation by the anammox community, a correlation analysis was performed. The results are shown in Figure 7.

Figure 7 reflects a weak positive relationship between the biofilm growth of both bioaugmented and indigenous microbial communities and the content of aluminum, potassium, and calcium in the carrier materials. There is a weak positive correlation between the removal of ammonium by the indigenous microbial community with the presence of silicon and calcium, and for a bioaugmented community—only with silicon. N-NO_2_/N-NH_4_ in the case of the indigenous microbial community has a strong positive correlation with magnesium, and one weaker with iron. For the bioaugmented microbial community, a strong positive correlation of N-NO_2_/N-NH_4_ with potassium and phosphorus, essential biogenic elements, was observed. It was shown that the addition of various metal cations has a positive effect on the vital activity of anammox bacteria: K^+^, Fe^2+^, Ca^2+^, and Mg^2+^ ions help anammox bacteria to remain active under conditions of high salinity. The mechanism is associated with the synthesis of an additional amount of the extracellular polymeric matrix of granules [86]. Positively charged cations can also bind to a negatively charged cell surface, thereby stimulating granulation [87]. Iron seems to be the most important cation, and a stimulating effect on granulation, biofilm growth, anammox activity, and total nitrogen removal by the bioreactor community was noted for both Fe^2+^ and Fe^3+^ [65,74,87]. In long-term experiments, the addition of iron cations increased the anammox activity by up to 20% compared to the control [88]. Fe^2+^ is needed for redox reactions and electron transfer reactions in metabolic processes [89]. At the same time, an excess concentration of cations can adversely affect the anammox community, inducing the synthesis of smaller granules and even a decrease in anammox activity [90,91]. For example, components such as K_2_O and P_2_O_5_ showed a low negative correlation with both ammonium and nitrite consumption by the indigenous anammox community (Figure 7).

A weak correlation was noted for the process of ammonium removal by the bioaugmented microbial community with the following elements in the composition of mineral carriers: magnesium oxide, aluminum, potassium, iron, and phosphorus. The inverse relationship between aluminum and phosphate compounds is due to the sedimentation process [92]. It is also known that prolonged exposure of the anammox community to aluminum oxide affects the activity of all three key enzymes of the anammox process and the level of the key functional gene of anammox bacteria HzsA [93].

In the case of clays, the high intensity of biofouling compared to more crystallized minerals can be explained by the presence of a large and mobile interplanar distance containing biogenic elements such as Fe, K, Mg, Ca, and P. The ability of bentonite clays and illite to strongly swell in water leads to a strong increase in the interplanar distance, which increases the availability of elements for microorganisms. In kaolin clays, this ability was not so pronounced; therefore, in this case, the presence of impurity biogenic elements in the structure was the decisive factor. The fouling of illite is primarily due to the presence of potassium and iron in it. The high fouling of clays may be associated with their ability to accumulate ammonium ions (which is especially true for montmorillonite clays and vermiculite [94]), reducing their negative effect on microorganisms at elevated concentrations.

Biofouling of the loams of the ChMZ and B2 aquifers was connected with their polymineral composition. For example, potassium feldspars and clays can serve as a source of potassium, while clays, amorphous iron oxides, and ferruginous minerals, such as siderite, can serve as a source of iron. The greater intensity of fouling of the ChMZ rock, compared to the B2 rock by anammox bacteria, can be associated with a high content of available iron and potassium from clay minerals and potassium feldspar in the latter. This creates an advantage for the growth of fast-growing non-anammox bacteria that compete with them. At the same time, the intensity of respiration and the total area of the biofilm on the B2 and ChMZ rocks were close to each other. In the bioaugmentation mode, due to the increased concentration of anammox bacteria, anammox bacteria developed on both samples according to the N-NO_2_/N-NH_4_ ratio.

Zeolite biofouling is due to its high sorption capacity for many elements (K, Fe, etc.) and ability to accumulate water-like clays. However, the activity of microbial processes on Z1 is stronger than on Z2. Most likely, this is due to the fact that the iron in Z2 is in a reduced form, which cannot be used under reducing conditions. Expanded clay used in the work is a product of the calcination of kaolin clays; that is, it may contain elements necessary for microorganisms, along with high porosity and the ability to retain water, which also favors the growth of the microbial community. Poor fouling of Q and S is associated with the dominance of quartz in their composition and the low content of biogenic elements in them.

Not in all cases, a high degree of biofouling of the carrier can be associated with the successful removal of oxidized and reduced forms of nitrogen. It is possible that the high area of fouling of some carriers creates conditions for competition between anammox bacteria and other groups of bacteria. In our opinion, the decrease in nitrogen consumption on some carriers with a high specific surface area (expanded clay, vermiculite, and zeolite-containing tripoli) can be explained by biofouling by more rapidly growing microorganisms compared to anammox bacteria.

Thus, for biofouling by anammox bacteria, it is not necessary to use a material with a high specific surface area, such as expanded clay. First of all, it is important to choose a material that (1) will contain the elements necessary for anammox bacteria, and (2) will not provide advantages for the rapid development of bacteria of other groups.

Overall, bioaugmentation of cold groundwater with a “warm” anammox community improved nitrogen removal rates and biofilm formation on most of the used mineral carriers. The observed lower anammox activity after bioaugmentation in the presence of some mineral carriers may be due to inhibition by some mineral components or to other possible factors associated with these mineral carriers, e.g., an unsuitable surface for biofilm formation. Another and/or additional explanation of this observation could be the combination of the cold shock factor for the “warm” bioaugmenting community and inhibition that led to the decay of the “warm” anammox community and the release of ammonium into the medium.

Despite the initial positive results obtained in this work, further research is needed to improve the process of sustainable bioremediation of polluted underground ecosystems. Great potential lies in using cheap organic substrates or organic carbon-rich wastewater to combine the partial denitrification process, which produces nitrite, and the anammox process, which converts nitrite and ammonium into harmless molecular nitrogen. The effectiveness of this approach for the treatment of wastewater containing both ammonium and nitrate nitrogen using the anammox process has been experimentally confirmed in numerous works [95,96,97,98,99], but it has not previously been shown to treat groundwater. The rate limiting of the anammox process due to low temperature can be overcome by the preliminary adaptation of a “warm” anammox community to cold temperatures before bioaugmentation into an aquifer. One of the possible ways to both accumulate a sufficient amount of anammox biomass for bioaugmentation and adapt a “warm” anammox community to cold is to use a pilot bioreactor near the bioaugmentation wells. The reactor should be maintained at a temperature of 15–20 °C as a compromise between the moderate anammox biomass growth rate and cold adaptation, and fed with a mineral medium with a gradual increase in the proportion of groundwater with the addition of ammonium nitrogen and some cheap sources of organic carbon (e.g., molasses) to ensure stable nitrite production from partial denitrification. The anammox biomass obtained in this way can be simultaneously adapted to hazardous compounds contained in polluted groundwater. However, it should be taken into account that the current methods of adaptation of a “warm” anammox community to low temperatures (10–15 °C) take months or even years. Techniques such as cold shock [100,101] or gradual acclimation and cold shocks [102] have been shown to be promising. A feasibility study of these methods may also be the focus of future work on sustainable bioremediation of cold polluted groundwater ecosystems through bioaugmentation.

## 5. Conclusions

The development of biofilms under conditions of oxidative and nitrate stress in low-temperature underground habitats is a necessary factor in the anammox process. The results of the work can be summarized in the following main conclusions:Quartz and kaolin clays of the Kantatskoye and Kamalinskoye deposits, bentonite clay of the Berezovskoye deposit, and zeolite of the Kholinskoye deposit can be used as components of a permeable barrier in the removal of ammonium and nitrite nitrogen in situ in the anammox process using an indigenous cold anammox community dominated by Ca. Scalindua.It can be assumed that the activity of the anammox process is determined by a number of factors, including the presence of the necessary trace elements in the material, the features of the surface morphology, as well as competition with other groups of organisms that can develop on materials at a faster rate.Bioaugmentation with a “warm” anammox community with a predominance of Ca. Kuenenia resulted in improved nitrogen removal on most carriers, including bentonite clay of the Dinozavrovoye deposit, loamy rock, and zeolite-containing tripoli, in addition to carriers that perform best with the indigenous anammox community. To further enhance the anammox process in a cold, nitrate-rich underground habitat, research is needed on the feasibility of coupled partial denitrification–anammox and the adaptation of a “warm” anammox community to low temperatures and hazardous compounds contained in polluted groundwater.Biofouling of loamy aquifers is due to their polymineral composition. For example, potassium feldspars and clays can serve as a source of potassium, while clays, amorphous iron oxides, and ferruginous minerals, such as siderite, can serve as a source of iron. It is important to note that biofouling by bacteria carrying out the anammox process on the ChMZ rock was quite successful, which suggests that under optimal conditions, the anammox process is possible in situ in the polluted aquifer.It can be assumed that when polymineral rocks are overgrown with anammox bacteria, competition with more rapidly developing bacteria can take place. Thus, for biofouling by anammox bacteria, it is not necessary to use a mineral carrier with a high area, such as expanded clay. However, it is necessary to select a carrier that will contain the elements necessary for anammox bacteria, while not providing advantages for the rapid development of bacteria of other groups.

It should be noted that, in addition to the removal of nitrogen, biofouling of mineral carriers in the permeable barrier system is important for the immobilization of metals in cases of multicomponent contamination. Thus, biofouling of rocks in aquifers leads to an increase in the immobilization of actinides [103], and biofouling of zeolite leads to an increase in the efficiency of the immobilization of heavy metals [5,104].

## Figures and Tables

**Figure 1 biology-11-01421-f001:**
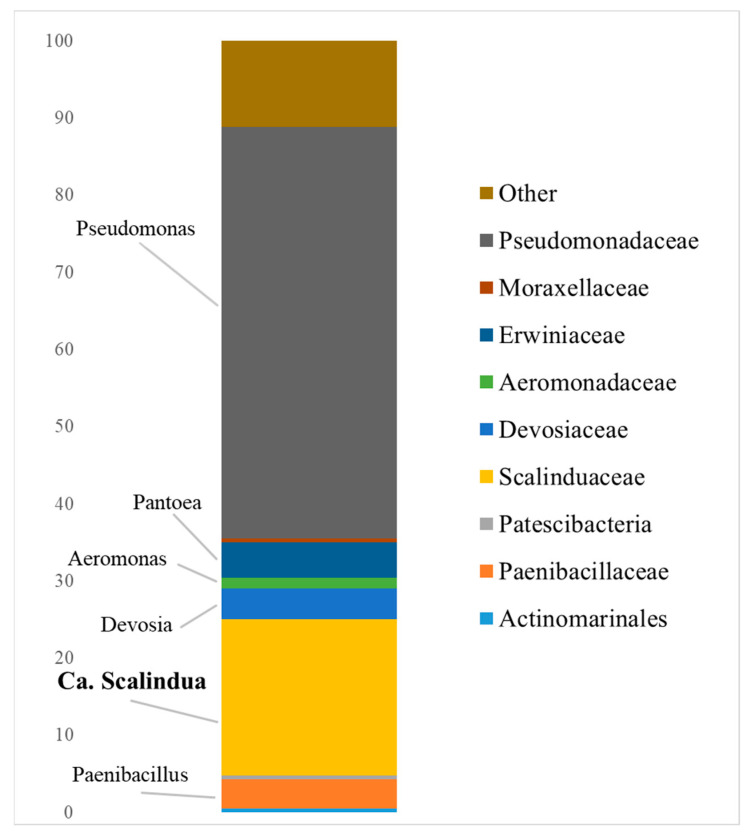
Taxonomic diversity of the microbial community from the water-bearing horizon on the family and genus level, based on 16S rRNA gene sequencing.

**Figure 2 biology-11-01421-f002:**
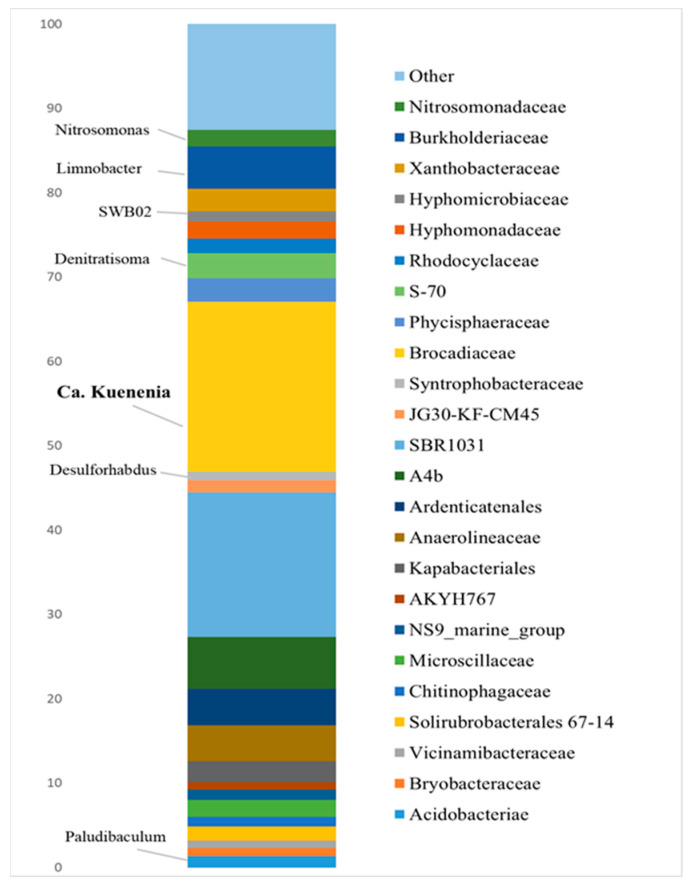
Taxonomic diversity of a “warm” anammox bioreactor community at the family and genus level, based on 16S rRNA gene sequencing.

**Figure 3 biology-11-01421-f003:**
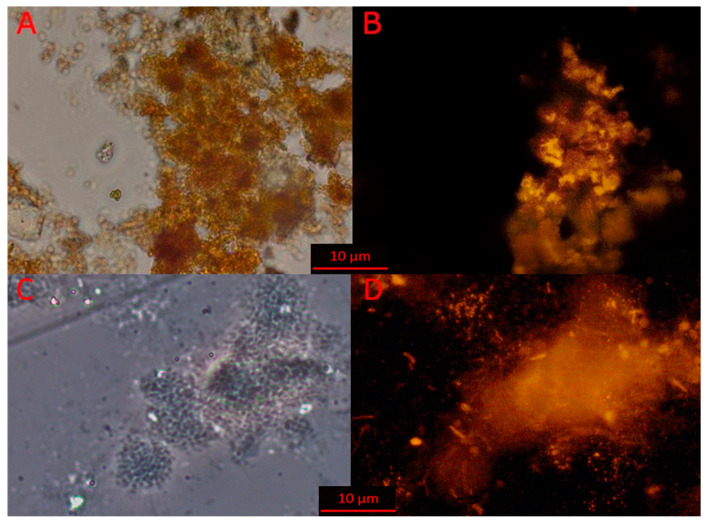
Phase-contrast (**A**,**C**) and fluorescent (**B**,**D**) microscopic images of microbial communities of groundwater (**A**,**B**) and a laboratory bioreactor (**C**,**D**). Scale bar 10 µm.

**Figure 4 biology-11-01421-f004:**
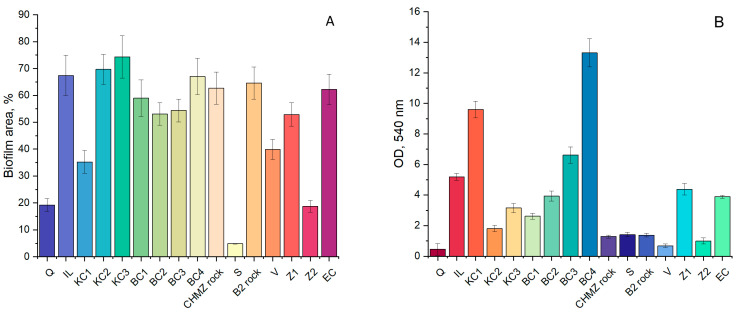
The area of biofouling of carriers by the indigenous microbial community, % according to confocal microscopy studies (**A**) and MTT assay (**B**).

**Figure 5 biology-11-01421-f005:**
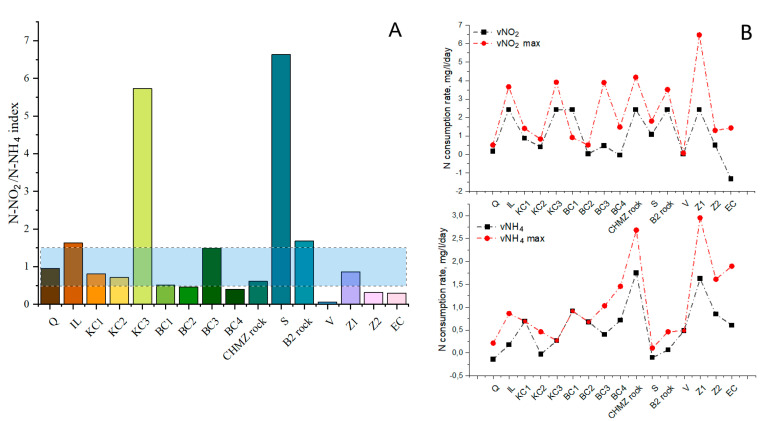
The ratio of consumed ammonium and nitrite nitrogen (N-NO_2_/N-NH_4_) (**A**) and ammonium and nitrite uptake rates (**B**) on mineral carriers by the indigenous microbial community.

**Figure 6 biology-11-01421-f006:**
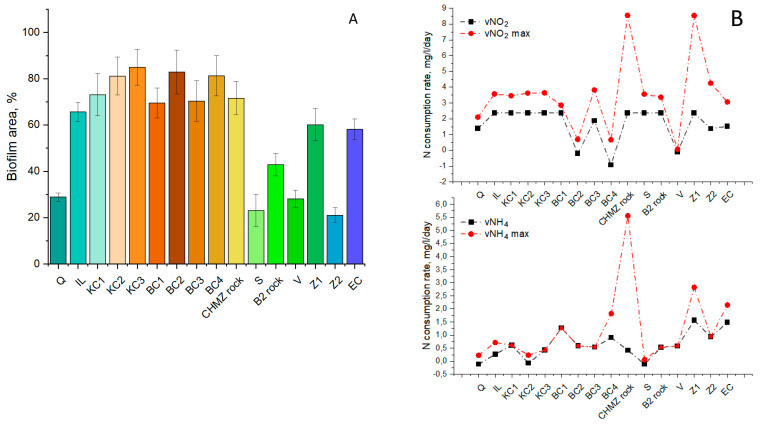
Biofilm area on carriers after bioaugmentation with a “warm” anammox community, % according to confocal microscopy (**A**) and rates of ammonium and nitrite consumption (**B**).

**Figure 7 biology-11-01421-f007:**
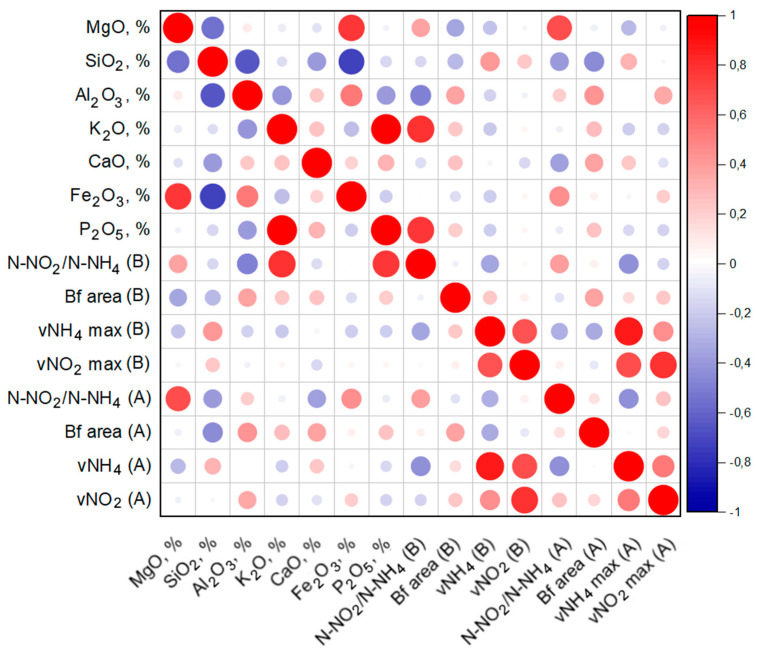
Correlation of N-NO_2_/N-NH_4_ index, nitrogen consumption, and biofilm growth (Bf area) with the elemental composition of mineral carriers, *p* ≤ 0.5. A—bioaugmented microbial community, B—indigenous microbial community.

**Table 1 biology-11-01421-t001:** Chemical composition of the polluted groundwater.

Parameter	Units	Value
NH_4_	mg/L	58.4
NO_2_	mg/L	28.18
NO_3_	mg/L	7434
HCO_3_	mg/L	244
K	mg/L	447.6
Na	mg/L	879.7
Mg	mg/L	37.6
Ca	mg/L	1310
Cl	mg/L	1213
SO_4_	mg/L	1803
Fe_tot_	mg/L	2.1
Mn	mg/L	15.7
PO_4_	mg/L	1.1
U	µg/L	256
C_org_	mg/L	2.5
pH	-	7.1
Eh	mV	119
T	°C	7.5

**Table 2 biology-11-01421-t002:** Mineral carriers used in this study.

Abbreviation	The Nature of the Carrier	Sampling Site
Q	Quartz	Kola Peninsula
V	Expanded vermiculite	Provided by PLC “Primver”
Z1	Zeolite (clinoptillolite)	Kholinskoye deposit, Khiloksky district, Trans-Baikal Territory
Z2	Zeolite-containing tripoli	Hotynetskoye deposit, Oryol region
EC	Expanded clay	CJSC “Keramzit” Moscow Region, Serpukhov district
IL	Illite	Ulyanovsk region
KC1	Kaolin clay	Kantatskiy deposit, Krasnoyarsk Territory
KC2	Kaolin clay	Kamalinskoye deposit, Abakan, Republic of Khakassia
KC3	Kaolin clay	Kompanovskoye deposit, Krasnoyarsk Territory
BC1	Bentonite clay	Dinozavrovoye deposit, Republic of Kazakhstan
BC2	Bentonite clay	Zyryanskoe deposit, Kurgan region
BC3	Bentonite clay	Berezovskoye deposit, Republic of Tatarstan
BC4	Bentonite clay	10 Hutor deposit, Republic of Khakassia
S	River sand	Chernogolovka (Moscow region)
B2 rock	Rock of the upper (11 m) water-bearing horizon	JSC “Siberian Chemical Plant” Tomsk region, ZATO Seversk
CHMZ rock	Rock of the water-bearing horizon	JSC “Chepetsky Mechanical Plant” Glazov, Republic of Udmurtia

**Table 3 biology-11-01421-t003:** Elemental composition of the mineral carriers, % mass.

Abbreviation	Na_2_O	MgO	Al_2_O_3_	SiO_2_	K_2_O	CaO	TiO_2_	MnO	Fe_2_O_3_	P_2_O_5_	S_tot._
Q	0	0	0	100	0	0	0	0	0	0	0
IL	0.23	1.57	12.22	68.1	2.73	0.55	0.8	0.02	4.42	0.05	0.11
KC1	0.68	1.11	19.71	62.0	1.3	1.81	1.01	0.04	4.54	0.08	<0.02
KC	0.3	0.53	21.9	61.8	2.72	0.3	0.57	0.04	2.06	0.06	<0.02
KC3	0.1	0.98	19.45	63.2	1.57	0.68	0.96	0.02	2.29	0.05	<0.02
BC1	1.07	3.33	15.58	53.0	0.06	1.45	0.53	0.24	4.63	0.02	0.05
BC2	0.57	1.73	16.22	49.7	0.94	3.93	0.8	0.07	5.94	0.04	0.08
BC3	0.36	2.46	18.54	55.8	2.34	1.27	0.89	0.05	7.77	0.14	0.33
BC4	0.91	2.73	16.94	56.2	0.85	2.05	0.69	0.03	3.36	0.12	0.04
S	0.46	0.17	2.55	94.9	0.6	0.22	0.08	0.003	0.44	0.03	<0.02
V	0.56	19.8	10.5	40.3	4.5	1.01	0.48	0.245	12.5	0.22	0.28
Z1	1.9	0.38	12.3	67.2	4	1.6	0.11	0.82	1	0.02	<0.02
Z2	0.1	1.4	8.5	70.1	1.4	2.1	0.54	0.14	4.3	0.22	<0.02
EC	0.86	2.2	18.8	60.1	3.8	3.4	1	0.152	8.2	0.29	0.1
B2 rock	1.96	0.72	10.23	81.1	2.09	1.55	0.52	0.044	2.25	0.07	<0.02
CHMZ rock	1.16	0.73	4.73	85.2	0.73	2.83	0.17	0.038	1.76	0.04	<0.02

**Table 4 biology-11-01421-t004:** Heat map of N-NO_2_/N-NH_4_ ratio and changes in biofouling and NH_4_ consumption rate after bioaugmentation with a “warm” anammox community.

Carrier Material	S *	NH_4_ *	N-NO_2_/N-NH_4_ *
Q	1.5	0.8	4.7
IL	1	1.5	3.4
KC1	2.1	0.9	1.5
KC2	1.2	2.8	12.5
KC3	1.1	1.6	2.1
BC1	1.2	1.4	0.8
BC2	1.6	0.9	0.2
BC3	1.3	1.3	1.3
BC4	1.2	1.3	0.4
CHMZ rock	1.1	0.2	1.2
S	4.7	1.1	8.4
B2 rock	0.7	7.7	1.3
V	0.7	1.2	0.1
Z1	1.1	1	0.6
Z2	1.1	1.1	0.6
EC	0.9	2.4	0.4

Note: *—The N-NO_2_/N-NH_4_ column in green shows N-NO_2_/N-NH_4_ values of 1 ± 0.5, which corresponds to the high contribution of the anammox process to nitrogen removal. In the rest of the columns, green means a significant improvement, yellow a slight improvement, and red a worsening.

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
