# Peer review of "Effect of Mineral Carriers on Biofilm Formation and Nitrogen Removal Activity by an Indigenous Anammox Community from Cold Groundwater Ecosystem Alone and Bioaugmented with Biomass from a “Warm” Anammox Reactor"

_biology, 2022, doi:10.3390/biology11101421_

Round 1
Reviewer 1 Report
In the Study, this purpose in was undertaken to evaluate the potential for bioremediation of nitrogen-polluted aquifers with indigenous anammox community adapted to a cold environment and survey the possibility of enhancing anaerobic ammonia oxidation activity in a cold subsurface environment. This article has positive implications for the study of cold anaerobic ammonia oxidation. However, there are still many parts that need to be improved in this paper. It was recommended for acceptance after major revision.
General Suggestion:
1. The conclusion needs to be appropriately streamlined. The results of this research can be summarised in points so that the reader can understand them at a glance.
2. The comparative analysis should be strengthened. Comparative analyses between two bacteria formed in cultures in different mineral media can be performed by increasing the conditions of comparison as appropriate to exclude the influence of other factors.
Specific suggestion:
1.Page 5, Table 2, “zeolite (clinoptillolite)” should be modified to “Zeolite (clinoptillolite)”.
2. Page 6, Table 3, The K2O column can be spaced appropriately to make the table more beautiful.
2. Page 8, Section 2.8, Some formulas can be added as appropriate to supplement the description and enrich the paragraph.
3. The authors should check and adjust these review references. Some recent research could be referenced, such as Bioresource Technology, 2022, 361, 127750; Bioresource Technology, 2022, 357, 127379; Journal of Environmental Management, 2021, 296, 113271; Chemosphere, 2021, 278, 130436.
4. Page 9, Figure 1, Ca. Scalindua is highlighted in bold and large type in the figure and not much more in the follow-up story.
5. Page 13, The title of Figure.4, “according to confocal microscopy studies (left) and MTT assay (right)” should be modified to “according to confocal microscopy studies (right) and MTT assay (left)”. Please check it.
6. Page 14, The right figure in Figure 5, “NO2” should be modified to “NO2”, “NH4” should be modified to “NH4”.
7. Page 16, The title of Figure.6, “according to confocal microscopy (left) and rates of ammonium and nitrite consumption (right)” should be modified to “according to confocal microscopy (right) and rates of ammonium and nitrite consumption (left)”. Please check it.
8. Page 17, Figure 7, some chemical formulas can be changed to the right format. For example, “Al2O3” be should modified to “Al2O3”. Please check them.
Author Response
In the Study, this purpose in was undertaken to evaluate the potential for bioremediation of nitrogen-polluted aquifers with indigenous anammox community adapted to a cold environment and survey the possibility of enhancing anaerobic ammonia oxidation activity in a cold subsurface environment. This article has positive implications for the study of cold anaerobic ammonia oxidation. However, there are still many parts that need to be improved in this paper. It was recommended for acceptance after major revision.
Response: Dear Reviewer, we thank you for the comprehensive and positive review of our manuscript. The comments were very constructive, and we tried to address all of the concerns. Below are the responses point by point.
General Suggestion:
- The conclusion needs to be appropriately streamlined. The results of this research can be summarised in points so that the reader can understand them at a glance.
Response: The conclusion section has been revised and summarized in points to better reflect the results of the work.
- The comparative analysis should be strengthened. Comparative analyses between two bacteria formed in cultures in different mineral media can be performed by increasing the conditions of comparison as appropriate to exclude the influence of other factors.
Response: To strengthen the work, a comparative analysis between the two (native and bioaugmentation) anammox microbial communities was added to the Discussion section.
Specific suggestion:
1.Page 5, Table 2, “zeolite (clinoptillolite)” should be modified to “Zeolite (clinoptillolite)”.
Response: The modification was made in accordance with the reviewer's suggestion.
- Page 6, Table 3, The K2O column can be spaced appropriately to make the table more beautiful.
Response: The K2O column was spaced appropriately in accordance with the reviewer's suggestion.
- Page 8, Section 2.8, Some formulas can be added as appropriate to supplement the description and enrich the paragraph.
Response: Equations were added.
- The authors should check and adjust these review references. Some recent research could be referenced, such as Bioresource Technology, 2022, 361, 127750; Bioresource Technology, 2022, 357, 127379; Journal of Environmental Management, 2021, 296, 113271; Chemosphere, 2021, 278, 130436.
Response: The papers suggested by the Reviewer have been cited at the appropriate places.
- Page 9, Figure 1, Ca. Scalindua is highlighted in bold and large type in the figure and not much more in the follow-up story.
Response: Ca. Scalindua in Fig.1 and Ca. Kuenenia in Fig.2 were indeed highlighted in bold and large type to show their belonging to anammox microorganisms. In addition, we have added more information about this important microbial group to the text.
- Page 13, The title of Figure.4, “according to confocal microscopy studies (left) and MTT assay (right)” should be modified to “according to confocal microscopy studies (right) and MTT assay (left)”. Please check it.
Response: Thank you, we have made the appropriate correction.
- Page 14, The right figure in Figure 5, “NO2” should bemodified to “NO2”, “NH4” should bemodified to “NH4”.
Response: Thank you, we have made the appropriate correction.
- Page 16, The title of Figure.6,“according to confocal microscopy (left) and rates of ammonium and nitrite consumption (right)” should be modified to “according to confocal microscopy (right) and rates of ammonium and nitrite consumption (left)”.Please check it.
Response: Thank you, we have made the appropriate correction.
- Page 17, Figure 7, some chemical formulas can be changed to the right format. For example, “Al2O3” be should modified to “Al2O3”.Please check them.
Response: Figure 7 has been corrected using subscripts where necessary.
Reviewer 2 Report
The research presented herein was undertaken to evaluate the potential for bioreme-diation of nitrogen-polluted aquifers with indigenous anammox community adapted to a cold environment. This topic is interesting, but some important details needs to be reassessed and/or elaborated further. The entire manuscript needs major revisions.
My questions and comments:
1. The whole study was conducted at 50 ml vials with the bioaugmenting anammox placed into one half of the vials. This means if the technoloy was applied, much anammox bacteria was needed. How do you ensure the enough supply of the warm anammox bacteria?
2. The anammox activity and growth after bioaugmentation was only characterized by the N-NO2/N-NH4 ratio and biofilm area. More information, such as QPCR or relative abundance characterized by High-throughput sequencing of 16S rRNA genes should be further supplied.
3. The groundwater contains much hazardous compounds. The anammox bacteria, espcially the bioaugmenting warm anammox should be sensitive to these hazardous compounds. How do the authors ensure the normal activity of the bioaugmenting anammox during the adaptation process, thus achieve a faster start-up?
4. The contribution of sulfate-reducing anaerobic ammonium oxidation should be further assessed. The bioaugmenting warm anammox should not be capable of ulfate-reducing anaerobic ammonium oxidation. If the sulfate-reducing anaerobic ammonium oxidation paly a major role in the groundwater, the strategy of bioaugmenting warm anammox to enhance the anammox activity may not effective.
5. The quality of Figures 3 was low, the authors should improve it.
Author Response
The research presented herein was undertaken to evaluate the potential for bioreme-diation of nitrogen-polluted aquifers with indigenous anammox community adapted to a cold environment. This topic is interesting, but some important details needs to be reassessed and/or elaborated further. The entire manuscript needs major revisions.
Response: Dear Reviewer, we thank you for the comprehensive and positive review of our manuscript. The comments were very constructive, and we tried to address all of the concerns. Below are the responses point by point.
My questions and comments:
- The whole study was conducted at 50 ml vials with the bioaugmenting anammox placed into one half of the vials. This means if the technoloy was applied, much anammox bacteria was needed. How do you ensure the enough supply of the warm anammox bacteria?
Response: Thank you for the good question. One of the possible ways to both accumulate a sufficient amount of biomass for bioaugmentation and adapt a warm anammox community to cold is to use a pilot bioreactor near the bioaugmentation well. The reactor should be maintained at 15-20°C as a compromise between moderate growth rate of anammox biomass and cold adaptation, and fed with groundwater supplemented with ammonium nitrogen and some cheap source of organic carbon (e.g. molasses) to ensure stable nitrite production as a result of partial denitrification.
We've added some discussion about this at the end of the Discussion section.
- The anammox activity and growth after bioaugmentation was only characterized by the N-NO2/N-NH4 ratio and biofilm area. More information, such as QPCR or relative abundance characterized by High-throughput sequencing of 16S rRNA genes should be further supplied.
Response: We thank the Reviewer for his valuable comment. One of the main goals of this work was the selection of mineral carriers for the best development of anammox bacteria, predominantly living in groundwater, in biofilms. We have tested a large number of carriers and identified the best. Indeed, we plan to continue experiments on a larger scale, first in a laboratory scale bioreactor and further in situ, in which we will use methods such as qPCR and High-throughput sequencing of 16S rRNA genes to assess the change in composition of microbial community and activity of anammox bacteria and other functional groups.
- The groundwater contains much hazardous compounds. The anammox bacteria, espcially the bioaugmenting warm anammox should be sensitive to these hazardous compounds. How do the authors ensure the normal activity of the bioaugmenting anammox during the adaptation process, thus achieve a faster start-up?
Response: We thank the reviewer for the question; indeed, this point must be taken into account in order to increase the success of bioaugmentation. As part of the response to the first reviewer's comment, we pointed out that in order to adapt warm anammox biomass, adaptation should be envisaged by gradually increasing the proportion of contaminated water in the mineral medium fed to the pilot reactor for the growth of bioaugmentation biomass.
We've added some discussion about this at the end of the Discussion section.
- The contribution of sulfate-reducing anaerobic ammonium oxidation should be further assessed. The bioaugmenting warm anammox should not be capable of ulfate-reducing anaerobic ammonium oxidation. If the sulfate-reducing anaerobic ammonium oxidation paly a major role in the groundwater, the strategy of bioaugmenting warm anammox to enhance the anammox activity may not effective.
Response: Thanks to the Reviewer for a deep analysis of the processes in this unique habitat. We will use this advice and carry out additional incubation experiments to assess the contribution of SRAO to the processes of nitrogen transformation by the microbial community. If the results obtained confirm the significant contribution of SRAO, further warm anammox bioaugmentation experiments will be adjusted to reflect new knowledge.
- The quality of Figures 3 was low, the authors should improve it.
Response: The quality of Figures 3 was improved.
Round 2
Reviewer 2 Report
No Suggestions.